# Analysis of PLA Geometric Properties Processed by FFF Additive Manufacturing: Effects of Process Parameters and Plate-Extruder Precision Motion

**DOI:** 10.3390/polym11101581

**Published:** 2019-09-27

**Authors:** Eustaquio García Plaza, Pedro José Núñez López, Miguel Ángel Caminero Torija, Jesús Miguel Chacón Muñoz

**Affiliations:** 1Energy Research and Industrial Applications Institute (INEI), Higher Technical School of Industrial Engineering, Department Applied Mechanics & Engineering of Projects, University of Castilla-La Mancha, Avda. Camilo José Cela, s/n, 13005 Ciudad Real, Spain; pedro.nunez@uclm.es (P.J.N.L.); miguelangel.caminero@uclm.es (M.Á.C.T.);; 2Higher Technical School of Industrial Engineering, IMACI, Department Applied Mechanics & Engineering of Projects, University of Castilla-La Mancha, Avda. Camilo José Cela, s/n, 13005 Ciudad Real, Spain

**Keywords:** Dimensional accuracy, Flatness, Surface texture, Fused filament fabrication (FFF), Polylactic Acid (PLA)

## Abstract

The evolution of fused filament fabrication (FFF) technology, initially restricted to the manufacturing of prototypes, has led to its application in the manufacture of finished functional products with excellent mechanical properties. However, FFF technology entails drawbacks in aspects, such as dimensional and geometric precision, and surface finish. These aspects are crucial for the assembly and service life of functional parts, with geometric qualities lagging far behind the optimum levels obtained by conventional manufacturing processes. A further shortcoming is the proliferation of low cost FFF 3D printers with low quality mechanical components, and malfunctions that have a critical impact on the quality of finished products. FFF product quality is directly influenced by printer settings, material properties in terms of cured layers, and the functional mechanical efficiency of the 3D printer. This paper analyzes the effect of the build orientation (*Bo*), layer thickness (*Lt*), feed rate (*Fr*) parameters, and plate-extruder movements on the dimensional accuracy, flatness error, and surface texture of polylactic acid (PLA) using a low cost open-source FFF 3D printer. The mathematical modelling of geometric properties was performed using artificial neural networks (ANN). The results showed that thinner layer thickness generated lower dimensional deviations, and feed rate had a minor influence on dimensional accuracy. The flatness error and surface texture showed a quasi-linear behavior correlated to layer thickness and feed rate, with alterations produced by 3D printer malfunctions. The mathematical models provide a comprehensive analysis of the geometric behavior of PLA processing by FFF, in order to identify optimum print settings for the processing of functional components.

## 1. Introduction

Fused filament fabrication (FFF) is a very promising technology for the processing of polymers and reinforced polymers, due to its versatility and simplicity in building all types of geometries using a wide range of materials. In recent years, the development of FFF has been exponential, with a technological evolution has enabled not only the manufacture of prototypes, but also of functional components with good mechanical properties. Notwithstanding, despite these improvements, crucial aspects, such as dimensional and geometric precision, or surface quality still lags far behind the qualities obtained by conventional manufacturing processes such as machining and forming, and impose serious limitations affecting the assembly and working life of manufactured components. 

FFF is one of the most used additive manufacturing processes owing to the simplicity of the processing procedures, where an extruded thermoplastic filament is deposited in individually superimposed layers to build a 3D geometry. FFF can be used with a broad range of thermoplastics, and their equivalent glass or carbon-fiber reinforced thermoplastics [1,2]. PLA polymer is one of the most extensively used materials in functional products, as it is biodegradable, non-contaminant material, exhibiting good mechanical properties that are frequently used in aerospace, automotive, and biomedical engineering applications [3,4,5]. 

The FFF additive manufacturing requires the configuration of process control parameters, such as layer thickness, feed rate, build orientation, raster pattern, raster angle, raster width, temperature, among others, that have direct repercussions of the efficiency of the process, and the quality of products. Correct parameter settings affect not only mechanical properties, as reported by Chacón et al. [6,7], but also dimensional precision, a geometric quality, and the surface texture obtained with this technology [8]. As the technology is relatively new, reference data on dimensional and geometric tolerances, and surface quality are not available. Moreover, the characteristics described by 3D printer manufacturers do not always match the results obtained [9,10], and with low cost FFF machines, this problem is further exacerbated. 

The optimum configuration of printing parameters in FFF additive manufacturing involves averting serious geometric deviations [11], undermining the quality del product. Thus, research in recent years has tended to focus on these aspects, in particular, dimensional accuracy, where most experimental studies have sought to characterize the influence of 3D printing parameters. Kumar et al. [12] and Sahu et al. [13] designed a predictive model of dimensional precision using as input parameters layer thickness, part orientation, raster angle, air gap, and raster width on ABS-P400. Chang et al. [14] examined the contour width effect, contour depth, part raster width, and raster angle on profile errors using image analysis of ABS material. Peng et al. [15] evaluated the accuracy and efficiency of the FFF process with ABS material using the parameters line width compensation, extrusion velocity, filling velocity, and layer thickness. Ahmed et al. [16] established a nonlinear relationship between dimensional accuracy and process parameters (layer thickness, air gap, raster angle, build orientation, road width, and number of contours). Kaveh et al. [17] examined the effects of the 3D printing parameters (extruded temperature, feed rate, flow rate, and raster width) on HIPS material. Rahman et al. [18] used the bed temperature, number of loops, nozzle temperature, print speed, layer thickness, and infill to evaluate dimensional precision, Noriega et al. [19] proposed a method for improving dimensional precision between parallel faces using ABS.

Other authors have analyzed the influence of other aspects related to the configuration of the 3D printers. Lee and Liu [20] analyzed the effect of the cooling air velocity in the dimensional quality of PLA material; higher cooling speeds generated better geometric accuracy, but were found lower mechanical strength. Galantuchi et al. [21] compared dimensional precision in an industrial versus a low cost machine, where severe shrinkage was observed to affect precision in the low cost machine negatively.

Occasionally, theoretical models have been built to calculate dimensional precision. Boschetto and Botini [22] developed a geometric model of the filament according to deposition angle and layer thickness to predict workpiece dimensions. Bosscheto et al. [23] broaden their study to include circular and spherical forms, and specific geometries, such as air fan blades. On the basis of the theoretical models obtained [22], Boschetto and Botini [24] proposed a method for working directly on solid CAD models, and for improving dimensional precision. Brenken et al. [25] developed a simulation tool named Additive3D to model the FFF process for fiber-reinforced thermoplastic composites. 

The characterization of form errors has received little attention in the literature, despite being a key aspect in quality control that affects the assembly and functioning of functional parts. These parameters are influenced not only by 3D printing parameters, but also by the dynamic behavior and malfunctions of the axes and spindle in FFF 3D printers. Roundness has been the most frequently analyzed parameter [23,26,27], whereas, research in other parameters, such as flatness has been comparatively scarce [26,28]. Sajan et al. [29] found that bed temperature, number of loops, nozzle temperature, print speed, layer thickness, and infill significantly influenced the roundness error on ABS. In Mahmood et al. [26], deviations were found to increase the size of geometric features, and deviations were greater for recessed features (holes), than for extruded features (bosses) in the X-Y plane. Reyes-Rodriguez et al. [27] analyzed dimensionally and form errors of polycarbonate (PC), where form deviations and dimensions depended mainly on orientation, whereas, nozzle diameter influenced the in-plane processing accuracy. Núñez et al. [28] determined that the best dimensional behavior was obtained with maximum layer thickness and solid density, and the minimum flatness error was obtained with less layer thickness and solid density. 

Similarly, surface finish, which has received little attention in comparison to dimensional accuracy, has been characterized. Ahn et al. [30] built a predictive model of surface finish, including factors, such as surface angle, layer thickness, cross-sectional shape of the filament, and overlap interval. Boscheto et al. [31] designed a similar theoretical model of surface finish to analyze layer thickness, tilting, model filling, support typology, and stratification angle on ABS material. A further study [32] examined other materials, such as ABS plus, Ultem 9085, and Polycarbonate. Boscheto et al. [33] proposed a new predictive model of finish surface to resolve the deficiencies observed in [31]. To complement the previously mentioned studies, Boschetto et al. [34] described a methodology for integrating surface finish predictive models in the CAD design process. Jin et al. [35] showed that good top surface finishes were obtained by coordinating the speed of filament driving motor and the axes driving motors synchronously. Reddy et al. [36] observed that roughness decreased with increased build inclination and increase with layer thickness, with no influence from the other parameters. Occasionally, to improve the surface roughness in FFF processes, heat treatments have been applied [37].

Currently, most studies published on the geometric characterization of polymer processing by FFF tend to focus on Acrylonitrile Butadiene Styrene (ABS) [38], a material widely used in FFF owing to its mechanical properties. In comparison, PLA, a non-contaminant and biodegradable polymer with good potential in the manufacture of functional products by FFF, has been scarcely researched in terms of geometric characterization. The manufacture of functional products requires dimensional and geometric tolerances that allow for the mass production and assembly of parts of a finished product. Moreover, surface quality is crucial for the functioning of moving components by decisively affecting tribological properties (wear, friction, lubrication, working life, etc.), and for optimizing their external visual aspect. This underscores the need for characterizing surface textures in order to determine optimum processing conditions.

In this study, the influence of the 3D printing parameters on the geometric behavior of PLA processed by FFF additive manufacturing was evaluated. The effect of build orientation (*Bo*) in the positioning of the object on the build plate, layer thickness (*Lt*), and feed rate (*Fr*) on the dimensional accuracy, flatness error, surface texture and roughness were analyzed. In addition, the effect of plate and extruder precision motion also were studied. Samples were manufactured using a low cost and open-source 3D printer, frequently used in the fields of science and technology. The methodology for data analysis was based on the statistical techniques of variance analysis, artificial neuronal network, and response surface. Finally, surface texture and roughness 3D topography images were evaluated to determine the effect of the process parameters and the plate-extruder precision motion.

## 2. Experimental Procedure 

PLA is a polymer amorphous semicrystalline with a high glass temperature (55–60 °C), very stable, ecological, recyclable, and odor free, which makes it an ideal material for the manufacturing of non-contaminant disposable functional components for a wide range of applications and sectors. PLA polymer is one of the most commonly used materials in 3D printing, as it is biodegradable with excellent properties for the application of FFF. PLA polymer processing exhibits strong adherence at ambient temperature, allowing for direct extrusion on the build plate using conventional methods. It solidifies rapidly, an important property when increased extrusion speed and reduced printing times are required. Moreover, PLA polymer has low thermal tension, which reduces warping, due to small thermal differences between working in a cold room (at ambient temperature), and the temperature of the workpiece material (190–220 °C). In this study a polylactide resin (Smart Materials 3D printing SL, Jaen, Spain) with reference 9051-89-2, was processed at 98% weight, 1.24 g/cm^3^ density, a fusion point of 145–160 °C, water insoluble, hardness of 64 Sh(D), and glass transition temperature of 56–64 °C. This polymer was obtained by polymerization of lactic acid derived from fermented vegetable sugars. The PLA polymer processing by FFF 3D printer melts in the extruder (hot-end) at 175 °C and above, though the recommended temperature is above 200 °C. According to the manufacturer, the temperature optimizing the printing speed/quality of this polymer is 205 °C. The PLA polymer filament is 1.75 mm in diameter, with a nozzle diameter of 0.4 mm. 

In order to evaluate the influence of processing parameters, build orientation (*Bo*), layer thickness (*Lt*), and feed rate (*Fr*) on the geometric properties of PLA polymer processing by FFF, a factorial design with three factors at different levels was performed: *Bo* was analyzed at three levels (upright, on-edge, flat), *Lt* was analyzed at four levels (0.06, 0.12, 0.18, 0.24 mm), and *Fr* was analyzed at three levels (20, 50, 80 mm/s). For the optimum characterization of the process, three replica workpieces were manufactured for each experimental condition, with a total of 3*Bo* × 4*Lt* × 3*Fr* × 3 = 108 test samples. The workpiece dimensions designed under the norm ISO 178 [39] were 80x10x4 mm (Figure 1). The layer fill was a solid fill density (100%), as this was expected to be the most appropriate density for functional components. The workpieces were manufactured on a low cost FFF 3D printer BQ Witbox (BQ, Madrid, Spain) with a build volume of 297 × 210 × 200 mm.

In order to determine the geometric behavior of PLA polymer processing by FFF additive manufacturing, three build orientations were evaluated: Upright (U), on-edge (O), and flat (F) (Figure 1). The manufacturing axes were determined according to the direction of the extruder: Longitudinal X-axis and transversal Y-axis (plane X-Y), and vertical Z-axis perpendicular to the build plate (Figure 1). The dimensional ratio between axes (A) was defined as the quotient between the workpiece dimension on the Z axis and the product of the dimensions on the X and Y axes. The three build orientations had values complementary to the coefficient A: A high value for the upright orientation (A = 1.625), an intermediate one for on-edge (A = 0.038), and a reduced value for the flat orientation (A = 0.006). The aim is to analyze how the longitudinal relations between axes influence the geometric precision of PLA polymer processing.

The dimensional deviations were measured by a coordinate measuring machine (CMM) Etalon Derby (Tesa Technology, Renens, Switzerland) with a 3D stylus Tesastar-i with a 2-mm diameter ball stylus, with axes ranges: X = 457 mm, Y = 508 mm and Z = 406 mm (Figure 2a). The resolution of the CMM was 0.001 mm, and the repeatability was defined by $E_{x,y,z}=0.004+0.005L/1000$, where *L* is the machine axis length. Sample dimensions were evaluated in the three manufacturing axes (X, Y, Z) as the distance between the sampling plane and the supporting surface. In position (1), 6 points were measured, due to the reduced dimensions of the sampling plane (Figure 2a). In positions (2) and (3), 10 points were taken in the sampling planes. The dimensional deviations on the three manufacturing axes $\left(\Delta D_{x},\Delta D_{y},\Delta D_{z}\right)$ were defined as the difference between the nominal value (theoretical value), as determined by the Cura software (Ultimaker B.V., Ultrecht, Netherlands), and the average of the experimental values (Equation It is valid. 1),
(1)$$\Delta D_{i}=D_{n_{i}}-\frac{1}{m}\sum_{j=1}^{m}D_{e_{j}},$$
where *i* is the corresponding axis (X, Y, or Z), $D_{ni}$ the theoretical value, $D_{ej}$ the experimental value at measurement point *j*, and *m* the number of point measurement (6 or 10, depending on the measurement position). 

The flatness, surface texture, and roughness were evaluated with a 3D surface profiling system Talysurf CLI-1000 It is valid. (Taylor-Hobson, Leicester, United Kingdom) equipped with an inductive contact gauge of Z-axis measurement range of 2.5 mm, with a resolution of 40 nm, and a measurement speed of 3 mm/s. Samplings were undertaken in the central part of the specimen face with the greatest surface area (Figure 2b), covering a 10 × 10 mm surface. The flatness deviation was evaluated using the *root mean square flatness deviation* (*fltq*) parameter [40], and the surface texture with the construction of 3D surface topography non-filtered. Surface roughness was evaluated using the parameters *arithmetic mean height* (*Sa*), *maximum height of peaks and valleys* (*Sz*) [41], and the 3D roughness topography using a Gaussian filter of *λ_c_* = 0.8 mm.

The *fltq* parameter (Equation (2)) is defined as the square root of the sum of the squares of the local flatness deviations from the *Least Squares Reference Plane* (LSPL) [40],
(2)$$fltq=\sqrt{\frac{1}{A}\int_{A}\Delta F_{1}^{2}dA},$$
where $\Delta F_{1}$ is the local flatness deviation, and *A* is the flatness area of the geometric element. For a 3D application, the parameter *fltq* was determined by the Equation (3),
(3)$$fltq=\sqrt{\frac{\sum {\left(m_{1}x+m_{2}y-2\right)}^{2}}{\left(n-2\right)\left(m_{1}^{2}+m_{2}^{2}+1\right)}},$$
where 

$m_{1}=\frac{\sum y^{2}\sum yz-\sum xy\sum yz}{{\sum x^{2}\sum y^{2}-\left(\sum xy\right)}^{2}}$, $m_{2}=\frac{\sum x^{2}\sum yz-\sum xy\sum yz}{{\sum x^{2}\sum y^{2}-\left(\sum xy\right)}^{2}}$, and $z=m_{1}x+m_{2}y$.

The *Sa* parameter (Equation (4)) quantifies the deviations in the height of the surface points in relation to the mean reference plane [41],
(4)$$Sa=\frac{1}{A}\iint_{A}\left|z(x,y)\right|dxdy,$$
where *A* is the measurement area.

The *Sz* parameter (Equation (5)) is defined as the sum of the largest peak height value and the largest pit depth value within the defined area [41],
(5)Sz=max(z(x,y))+min(z(x,y)).

## 3. Results and Discussion

### 3.1. Dimensional Characterization

Dimensional accuracy was evaluated independently in the three orthogonal axes of the movement of the extruder (X, Y, Z) differentiating the contour directions and the filling layer (plane X-Y), and the direction perpendicular to the layer (Z) in each workpiece (Figure 2a). This allows for the analysis of dimensional deviations produced by the precision positioning of the extruder in producing each layer and the cured filament, and the dimensional deviations with the accumulation of layers, which was influenced by the movement of the build plate and the cured composite materials layer by layer. Table 1 shows the analysis of variance (ANOVA) for a second-order model showing the influence of the feed rate (*Fr),* and layer thickness *(Lt)* parameters on dimensional deviations (*ΔD_i_*) in the build orientation analyzed. The results showed that the two parameters examined (*Fr* and *Lt*) were not significant in any of the build orientations, with *p*-values greater than 0.05. This implies that neither of the independent variables (*Fr* and *Lt*) had an influence on the dependent variable (*ΔD_i_),* or that there is no linear behavior between the input and output variables. 

In the analysis of the linearity between input (*Fr*, *Lt*) and output (*ΔD_x_*, *ΔD_y_*, *ΔD_z_*) using Pearson’s test, the results of Table 2 show low linearity between input and output, implying inadequate modelling of experimental data using regression techniques [12]. As an alternative to regression models, artificial neural networks (ANN) were applied to model the behavior of dimensional deviations. In manufacturing processes, ANNs have been frequently used to estimate a broad range of process parameters, i.e., tool wear, dimensional accuracy, surface roughness, etc. For ANN to provide good results, several factors must be selected adequately. Most of the studies apply the *feedforward backpropagation* method [12,42], due to its computational efficiency. In relation to the training function, two main functions have been used, the *Levenberg-Marquardt* (LM) and *Bayesian Regularization* (BR) [42]. Furthermore, there are several transference functions, being the *hyperbolic tangent* and the *linear function* the most used [42]. As for the neural networks structure, most of the authors agree to the lower hidden layers and neurons is the best option. Thus, ANNs based on *feedforward backpropagation* were optimized, and the best network results were obtained with the LM training function, the *hyperbolic tangent* transference function, and of a 2–8–4–1 network structure, selecting the optimum configuration with the lowest root mean square error (RMSE). Once the mathematical model was obtained, surface response (Figure 3) was represented individually for each direction measured (*X*, *Y*, *Z*), and build orientation (*U*, *O*, *F*), and for each case the corresponding evolution of dimensional deviations (*ΔD_x_*, *ΔD_y_*, *ΔD_z_*), and their tolerances according to the *Fr* and *Lt* processing parameters. 

Figure 3 shows surface responses obtained using artificial neural networks built on the experimental data. The analysis of dimensional behavior on the *X*-axis, the upright (Figure 3a), and on-edge (Figure 3d) building orientation revealed a very similar behavior, with a maximum deviation of 140 µm and a minimum of −39 µm for the upright position, and a maximum of 170 µm and a minimum of −38 µm for the on-edge position. The mean dimensional variability between samples was low, ±18 µm and ±21 µm, respectively. In contrast, in the flat position (Figure 3g), significant deviations were found in the nominal value with a maximum deviation of 700 µm and a minimum of 262 µm, and a mean variability of ±84 µm. As shown in Figure 4, this phenomenon occurs, due to deficiently cured material in the central layers of the workpiece, the effects being directly proportional to layer thickness, that is, the greater the layer thickness, the greater the dimensional error. In most cases, the smallest layer thickness of 0.06 mm had the lowest dimensional deviations (*ΔD_x_*) in all three build orientations (Figure 3a,d,g). The optimum deviations were obtained for a feed rate of 20 mm/s to 50 mm/s, and a layer thickness of 0.06 mm, with deviations of −7 ± 47 µm for the upright, 7 ± 58 µm for the on-edge, and 262 ± 107 µm for the flat building orientation.

In comparison, the behavior of the *Y*-axis was substantially different in all three build orientations. In the upright position (Figure 3b), dimensional deviations were high in all of the experimental conditions, with a maximum deviation of 525 µm and a minimum of 339 µm, and a mean variability of ±79 µm. These high dimensional deviations were produced by the excessive fused filament length in the central area of each layer, which is where extrusion begins. This phenomenon is shown in Figure 5a for the configuration *Lt* = 0.24 mm and *Fr* = 50 mm/s, where the length of the two fused central filaments was longer than the nominal, which produced a systematic deviation in the *Y*-axis in all of the workpieces. The smallest dimensional deviations were in *Fr* = 80 mm/s and *Lt* = 0.12 mm with deviations of 339 ± 170 µm. For the on-edge position (Figure 3e), a highly irregular behavior with negative dimensional deviations in all of the configurations was observed. A maximum of -10 µm and a minimum of −350 µm, with a mean variability of ±97 µm were obtained. As in the upright position, these dimensional deviations were produced by deficiencies in the two fused central filaments, but in this position, it was due to shortening of the length. This effect is shown in Figure 5b, for the highest (*Lt* = 0.12 mm and *Fr* = 20 mm/s), and lowest conditions (*Lt* = 0.24 mm y *Fr* = 50 mm/s) of dimensional deviation, with a shortening of the central filament in relation to the nominal value of 76.440 mm and 76.802 mm, respectively. In this case, feed rate influenced dimensional behavior with a reduction in deviations for all *Fr* = 50 mm/s, the optimum being *Fr* = 50 mm/s and *Lt* = 0.24 mm with deviations of −10 ± 95 µm. In the flat position (Figure 3h), dimensional deviations increased with increasing layer thickness, with no significant influence of feed rate. A maximum deviation of 300 µm and a minimum of −183 µm was obtained, with a mean variability of ±70 µm. As in the previous cases, the deviations on the Y-axis for the flat position, produced by errors in the length of the fused central filament, obtained the lowest dimensional deviation for *Fr* = 20 mm/s and *Lt* = 0.06 mm of −88 ± 200 µm.

The best dimensional behavior was observed on the Z-axis (*ΔD_z_*) in the three build orientations. In the upright position (Figure 3c) obtained a maximum of 260 µm and a minimum of −12 µm, with a mean variability of ±61 µm. A moderate reduction in deviations occurred with an increased layer thickness until they reached *Lt* = 0.18, where the deviations were minimum and stabilized with values very close to nominal ones. Excellent results were obtained in dimensional precision for layer thickness values ranging from 0.18 mm to 0.24 mm in all of the feed rates analyzed, with an optimum configuration of *Fr* = 20 mm/s and *Lt* = 0.24 mm, with an 8 ± 109 µm deviation. The on-edge build orientation (Figure 3f) also obtained good results, with a maximum of 178 µm and a minimum of −134 µm, and variability of ±65 µm. In this case, the feed rate slightly influenced *ΔD_z_*, mainly in the smaller layer thickness of 0.06 mm, where an increase in *Fr* increased dimensional deviation, the optimum configuration being *Fr* = 20 mm/s and *Lt* = 0.06 mm with a deviation of 13 ± 140 µm. In the flat position (Figure 3i), all of the deviations contained negative values versus the nominal ones, with a maximum of −101 µm and a minimum of −228 µm, and a mean variability of ±25 µm. In this position, the best results were obtained for the smallest layer thickness of 0.06 mm in all the ranges of feed rate examined. The deviations slight increased, only in *Lt* = 0.12 mm, with the optimum configuration being *Fr* = 20 mm/s and *Lt* = 0.06 mm, and a deviation of −101 ± 43 µm.

As for the absolute error, as determined by the difference between the nominal value, and the average of the experimental values (Equation (1)), the X and Z axes showed the best dimensional behavior in the on-edge and upright positions with deviations below ~13 µm in comparison to nominal values. For the Y axis good results were only obtained in the on-edge orientation, with shorter movements of the extrude, and an absolute dimensional error of ~10 µm, but bad results in the other orientations, with minimum errors ranging from ~90 µm to ~339 µm. In terms of build orientations, only the on-edge position obtained good dimensional results in the three axes, with deviations below ~13 µm. The flat orientation exhibited the worst behavior in all of the axes, with minimum dimensional errors ranging from ~90 µm to ~260 µm. The upright position worked well on the X and Z axes, with deviations below ~10 µm, but with deficient behavior of the Y axis (*ΔD_z_* = 339 µm). The optimum dimensional behavior was obtained with a feed rate of 20 and 50 mm/s, and in most cases, a layer thickness of 0.06 mm and 0.12 mm.

### 3.2. Flatness Evaluation

Flatness evaluation requires a broad surface in order to detect geometric changes to the reference plane [40]. Thus, the largest surface area was evaluated, corresponding to the 80 × 10 mm face. This enabled the evaluation of different aspects of 3D printing according to the build orientation of the workpiece: (a) In the position flat, flatness was evaluated in the upper layer that is affected by the X-Y trajectories of the extruder, the curing of the filament, and the small accumulation of layers; (b) in the upright position, flatness was evaluated in the accumulation of layers in a 8:1 ratio versus the on-edge position, and ranged from Z = 35 mm to Z = 45 mm, where there was already a substantial accumulation of layers, and the build orientation was very unstable; and (c) in the on-edge position, the accumulation of layers was evaluated from the height Z = 0 mm to Z = 10 mm, with greater stability in the build orientation owing to the fewer number of accumulated layers, but longer layer lengths of an 8:1 ratio versus the upright position on the X axis. Despite upright and on-edge positions flatness were evaluated in an area of accumulated layers, the upright position was more unstable and accumulated more layers, and a greater area of the height of the workpiece was evaluated. In the on-edge position, the stability and the surface contact with the build tray are greater, with smaller workpiece heights.

Table 3 shows the influence of process parameters (*Fr* and *Lt*) on flatness (*Fltq*) according to the second-order ANOVA analysis. This analysis was carried out separately for each build orientation, and the influence of each was evaluated using the type III sum of squares and the *p-values* for a 95% confidence interval. As shown in Table 3, all of the process parameters in the *upright* position and their interactions were significant on flatness (*Fltq*), with *p-values* below 0.05, except for the term *Lt^2^*. The *Lt* layer thickness had the highest sum of squares, indicating it was the most influential parameter. In the on-edge position, none of the machining parameters was significant, but *Lt* had a p-value close to 0.05, which could imply the influence of this parameter. The flat position obtained similar results as the on-edge position, none of the machining parameters had a significant influence on flatness, and all had very high *p-values*. 

To confirm the ANOVA analysis (Table 3), a response surface (Figure 6) was obtained from a 2–8–4 neuronal network structure. The evolution of flatness (*Fltq*) according to the *Fr* and *Lt* printing parameters for the three build orientations is shown in Figure 6. For the upright position (Figure 6a), the results obtained confirmed the ANOVA analysis (Table 3), where a strong influence of the *Fr* and *Lt* parameters on flatness was observed. For slower feed rates of 20 mm/s, a stable behavior was observed, without remarkable differences in any of the layer thicknesses evaluated, and flatness values ranging from 19.8 µm to 26.6 µm. In contrast, a strong interaction between the *Fr* and *Lt* parameters was found with increased feed rate, in particular, a feed rate of 80 mm/s. A 50 mm/s feed rate produced a sharp decline in flatness in the smallest layer thickness of 0.06 mm, with a minimum absolute *Fltq* of 7.8 µm. Moreover, in the 50 mm/s feed rate, an important increase in flatness was found in relation to increased layer thickness, with an increase in *Fltq* from 7.8 µm (*Lt* = 0.06 mm) to 31.9 µm (*Lt* = 0.24 mm). For the maximum feed rate of 80 mm/s, flatness increased sharply with increased layer thickness, an increase of *Fltq* from 16.9 µm (*Lt* = 0.06 mm) to 55.7 µm (*Lt* = 0.18 mm), which produced the absolute maximum. In the on-edge position (Figure 6b), the behavior was uniform in all the experimental combinations, without any significant pattern of behavior, and *Fltq* values ranged from 14.6 µm to 30.3 µm. A slight increase in flatness was observed in relation to increased layer thickness, which confirmed the results of the ANOVA analysis (Table 3), where *Lt* appeared to influence flatness. The optimum flatness was obtained with *Fr* = 50 mm/s and *Lt* = 0.06 mm, and an *Fltq* of 14.6 µm. In the flat orientation (Figure 6c), an irregular flatness behavior was observed without any pattern of behavior in relation to processing parameters, with *Fltq* ranging from 15.9 µm to 39.7 µm. This behavior revealed the low linearity of the data, which would explain that *Fr* and *Lt* processing parameters showed no significant influence on the ANOVA analysis (Table 3). The optimum flatness obtained was *Fr* = 50 mm/s and *Lt* = 0.18 mm with a *Fltq* of 15.9 µm.

The microgeometry obtained in the upright orientation for all of the analyzed layer thicknesses and feed rate extrusion is shown in Figure 7. Two characteristic phenomena were observed to affect the flatness of the evaluated surfaces differently. Firstly, the “bathtub ring” effect generates surfaces with very high deviations on the borders in comparison to the internal area of the workpiece. Secondly, transversal waviness observed in the direction perpendicular to the direction of extrusion. The “bathtub ring” effect depends directly on the feed rate and layer thickness, given that it does not occur at a low feed rate (20 mm/s). However, increasing the feed rate to 50 mm/s leads to a slight increase in the height of the borders, from *Lt* = 0.12 mm onwards (Figure 7e), and increases as *Lt* layer height increases. This effect was highest for a feed rate of 80 mm/s and a layer height of 0.24 mm (Figure 7l), with differences regarding a height of around ~250 µm between the borders and the central surface. The effect increased in accordance with layer thickness, with the highest deviations observed in *Lt* ranging from 0.18 to 0.24 mm. The “bathtub ring” effect was responsible for the significant increase in the flatness error (*Fltq*) at high feed rate and layer thicknesses (Figure 6), and was a product of excess material released by the extruder during the acceleration and deceleration phase of each trajectory.

This phenomenon was confirmed by a transversal section of the surface, as shown in Figure 8, obtained with the *Fr* = 80 mm/s, and *Lt* = 0.24 mm process parameters. As shown, the linear trajectory of the extruder consists of three different phases: In the first phase, the extruder accelerates abruptly, going from the still standby position (*Fr* = 0 mm/s) to the programmed feed rate (*Fr* = 80 m/s). In this phase, the extruder fails to take into account the transitory acceleration period, and injects the same quantity of material as when the speed is in continuous movement, leading to the deposition of excess material, given that real speed is slower than programmed speed in the G-code. In the second phase, the extruder injects the filament at a constant speed, producing a uniform layer of material without excess material. Finally, in the third phase, the extruder begins to decelerate until it comes to a halt in standby mode, leading to differences between the real speed and the programmed speed, which successively produce excess material in this area.

The second phenomenon examined was transversal waviness that appeared perpendicular to the direction of the extrusion. As shown in Figure 7, waviness appeared in all of the workpieces analyzed, and its effect increased with an increased layer thickness. At the highest feed rate (80 mm/s) and layer thickness (Figure 7i,l), waviness was less evident owing to the bathtub ring effect of these surfaces was more significant and minimized the effect. This transversal waviness was due to the cylindricity defect in the threaded spindle responsible for moving vertically (the Z-axis) the build plate. In order to corroborate this phenomenon, Figure 9 shows four profiles extracted from the surfaces evaluated from a perpendicular section to the direction of filament deposition, with a feed rate of 20 mm/s. As shown in Figure 9, all of the profiles displayed characteristic transversal waviness with a *T* period of ~1.92 mm, which would indicate this phenomenon was due to a periodic type of deviation in spindle rotation. The results showed increased layer thickness, diminished the number of layers per period in proportion to the size of the layer according to the expression *N*° *Layers* ≈ *T*/*Lt*. Thus, approximately 32 layers were obtained per period for the smallest layer thickness of 0.06 mm, that is, 32 layers per complete spindle rotation (Figure 9a). As layer thickness increased, the number of layers per period diminished proportionally, with 16 layers for *Lt* = 0.12 mm (Figure 9b), 12 layers for *Lt* = 0.18 mm (Figure 9c), and 8 layers for *Lt* = 0.24 mm (Figure 9d). 

For the on-edge and upright building orientations, only a case of a 3D topography image was analyzed, due to the non-significant influence of either the *Lt* or *Fr* processing parameters. Figure 10 shows the surface microgeometries obtained in both building orientations with a feed rate of 20 mm/s, and a layer thickness of 0.06 mm. As expected, the phenomenon of transversal waviness reappeared perpendicular to the deposition of layers in the on-edge orientation (Figure 10a), producing a negative effect on the flatness obtained. Figure 10b shows the 3D topography in the flat orientation, with the deposition of filaments in the final layer. It can be observed that the transversal waviness, owing to spindle rotation in the vertical direction, disappeared in this case, given that the build plate lacks any movement on the Z-axis. However, other deviations and irregularities with an important effect on flatness were observed. To illustrate these defects, Figure 11 shows a transversal section of the surface of Figure 10b produced by a plane perpendicular to the grooves. Firstly, In the longitudinal section (Figure 11a), an irregular waviness component was observed without a defined *T* period, and a high amplitude produced variations regarding a height of ~150 µm. This deviation was due to defective movement of the extruder on the longitudinal guides in the *X* direction. Likewise, the transversal section (Figure 11b) also showed a low frequency and low amplitude waviness component without a defined *T* period, producing deviations of ~17 µm, arising from deviations in the movement on the guides of the extruder in the *Y* direction.

### 3.3. Surface Roughness

Surface roughness was evaluated separately in each of the three building orientations in order to assess the effect of the *Lt* and *Fr* parameters on the *Sa* and *Sz* parameters [41]. The ANOVA analysis performed according to the build orientation is shown in Table 4. In the upright and on-edge orientations, *Lt* had a significant effect on the *Sa* and *Sz* parameters, with *p*-values always below 0.05, and a high of a sum of squares. However, *p*-values for *Fr* were above 0.05, and the sum of squares low, indicating no effect on either the *Sa* or *Sz* parameter. In contrast, the behavior in the flat build orientation was completely different, neither *Lt* nor *Fr* had a significant effect on *Sa* or *Sz*, with *p*-values above 0.05, and a low sum of squares.

To confirm the ANOVA analysis of Table 4, response surfaces were obtained (Figure 12) using a 2-8-4 artificial neuronal network structure. The evolution of both surface finish parameters *Sa* and *Sz* in relation to *Lt* and *Fr* factors for the three build orientations analyzed is shown in Figure 12. The *Sa* parameter, a similar behavior was observed in the upright and on-edge build orientation (Fig, 12a and 12b), with a linear growth according to layer thickness, but with no significant influence of feed rate, which confirmed the ANOVA analysis of Table 4. The best results were obtained for a layer thickness of 0.06 mm, with minimum *Sa* roughness values of 2.8 µm for the upright, and 2.5 µm for the on-edge orientation. Increased layer thickness, sharply increased *Sa* values, with a layer thickness of 0.24 mm, the maximum *Sa* values were 15.3 µm for the upright, and 15.6 µm for the on-edge orientation. For the flat position (Figure 12c) the *Sa* values obtained were higher than those obtained for the upright and on-edge positions, without significant influence of the *Fr* or *Lt* processing parameters, which confirmed the ANOVA analysis of Table 4. A maximum of 28.7 µm and a minimum of 25.5 µm were obtained for *Lt* = 0.18 mm, and *Fr* = 20 mm/s and Fr = 50 mm/s, respectively. In the flat orientation, surface roughness depends on the deposition and curing of the filament, whereas, in the upright and on-edge position, it depends on the texture generated by the accumulation of layers. 

The behavior of the *Sz* parameter was similar in the upright and on-edge positions (Figure 12d,e), with maximum values of 78.0 µm and 75.4 µm, and minimum values of 34.4 µm and 34.5 µm, respectively. A small quasi-linear growth of *Sz* was observed with an increased layer thickness in both build orientations, without the significant influence of feed rate. In the upright position (Figure 12d) a slight reduction in *Sz* was observed with increased feed rate for *Lt* = 0.06 mm, but this behavior did not remain constant in the other layer thicknesses, with a minimum value for *Fr* = 80 mm/s and *Lt* = 0.06 mm. In the on-edge position (Figure 12e), the minimum of *Sz* were *Fr* = 50 mm/s and *Lt* = 0.06 mm. In these build orientations, the minimum *Sz* values were for *Lt* = 0.06 mm, where the surfaces consisting of accumulated layers were evaluated. As in the analysis of the *Sa* parameter, the *Sz* values in the flat position (Figure 12f) were much higher than in the upright or on-edge positions, with a maximum of 171 µm, and a minimum of 131.8 µm. As can be observed, a greater variation of *Sz* was observed in this build orientation, but with no significant influence of the *Fr* or *Lt* parameters. The lowest *Sz* values were obtained for 50 ≤ *Fr* ≤ 80 mm/s, and 0.12 ≤ *Lt* ≤ 0.18 mm, reaching an absolute minimum for *Lt* = 0.18 mm and *Fr* = 50 mm/s. Once again, the results obtained for the *Sz* parameter confirmed the results obtained in the ANOVA analysis (Table 4).

The surface roughness obtained in the three build orientations is shown in Figure 13. For the upright position (Figure 13a–d), and on-edge position (Figure 13e–g and Figure 12h), where only the *Lt* parameter was significant (Table 4), the 3D surface roughness is displayed according to layer thickness, for a feed rate of 20 mm/s. In comparison, in the flat orientation (Figure 13i), where neither of the process parameters *Fr* or *Lt* were significant, only one single experimental condition is shown for *Fr* = 20 mm/s and *Lt* = 0.06 mm. The surface roughness was obtained by applying a Gaussian filter with a cut-off *λc* of 0.8 mm. 

In the upright orientation (Figure 13a–d), increased layer thickness from 0.06 mm to 0.24 mm increased peak heights from ~50 µm to ~70 µm. This was regardless of surface *Lt* = 0.06 mm (Figure 13a) having a maximum peak height of ~50 µm, whereas, surface *Lt* = 0.12 mm (Figure 13b) was ~40 µm. This phenomenon occurred because the height scale defined in the color bar, fixes the initial value in the deepest valley of the surface. As shown in Figure 13a, there were deep valleys in specific workpiece areas that were not representative of general surface behavior, but directly affected the maximum height values. As can be observed, these valleys were generated by the transversal waviness in the direction of extrusion; and a phenomenon referred to in a previous section (Figure 11). Waviness mainly affected surface roughness with very small layer thicknesses, such as *Lt* = 0.06 mm, whereas, with increased layer thickness, the greater thickness of the extruded filament had a greater effect on surface roughness, and minimized the waviness effect. In the on-edge build orientation (Figure 13–h), once again increased layer thickness progressively increased peak heights, with values ranging from ~50 µm to ~80 µm. In contrast to the upright build orientation, in the on-edge build orientation, there was homogeneous layer distribution in all four surfaces represented, with no transversal waviness was due to greater stability in the manufacturing of the samples, owing to the greater support surface of the build plate, and smaller heights that avoid oscillations. The flat orientation (Figure 13i) obtained the worst surface roughness values with peak heights reaching ~140 µm, as in this position the diameter of the extruded filament (0.4 mm) conditioned surface roughness, with a value much higher than in the layer thickness defining roughness in the upright and on-edge positions. In the extrusion direction, the filament deposited exhibited an irregular behavior produced by malformations in the extrusion of the material. 

### 3.4. Comparison

To our knowledge, no study has been published to date on the application of FFF process over PLA material analysing together the build orientation, layer thickness, and feed rate process parameters. Thus, an exhaustive comparative with other works was not possible owing to the different parameters and materials analyzed. Nevertheless, there are a few similarities with other works that allow us to draw a comparison focusing only on partial results. 

As we mentioned in the introductory section, the effect of process parameters on dimensional accuracy has been studied extensively using mainly ABS material. There is little consistency in works published in this field, with significant differences in the main conclusions. In the study of Kumar et al. [12], as in the present study, layer thickness and build orientation were found to be the process parameter most influencing dimensional accuracy. In contrast to our results, however, they observed shrinkage along the length and width of the workpiece, and positive deviation in the thickness. In the work published by Ahmed et al. [16], once again, found layer thickness and build orientation were the most significant process parameters with high values of the sum of squares in the ANOVA. The authors also observed an increase in length, width, and thickness with increasing layer thickness. For length and width, these results agreed with our results in the build orientation 0°. For thickness, the results obtained in our study were in agreement for 90°. In the study [16] the effect of build orientation on dimensional accuracy was also determined, and an increase in length and a decrease in width and thickness were found when the angle increased (0°–90°). These results were in accordance with our results for length at lower layer thicknesses, and for width at higher layer thicknesses. In the study with ABS material by Rahman et al. [18] a positive deviation in the Z-axis and shrinkage in the X-axis, and Y-axis were found, which is in opposition to our results. Galantuchi et al. [21] found that dimensional precision improved with reduced path speed, which is again in opposition to our results, where speed had no significant effect on dimensional accuracy. In the study developed by Reyes-Rodriguez et al. [27], build orientation was determined to be one of the most important parameters affecting dimensional errors and form deviation, which is in accordance with our results, especially regarding flatness deviation. The lack of consistency in some of our results with the findings of other studies are a largely a product of the type of material as to the properties of PLA and ABS are totally different.

In contrast, the results obtained for dimensional accuracy in this work related to surface finish are in accordance with most of the published works. Boscheto et al. [31,32,33,34] and Reddy et al. [36] obtained similar conclusions as in our work, where layer thickness was the most significant parameter in negatively affecting surface finish. 

## 4. Conclusions

This study analyzed the geometric properties of PLA polymer processing by FFF additive manufacturing at a low cost and open-source 3D printer. The effects of the processing parameters build orientation, layer thickness, and feed rate on dimensional accuracy, flatness error, and surface texture were characterized. The results showed a non-linear behavior of the experimental data, and the need for modelling geometric behavior using artificial neuronal networks. 

The build orientation and layer thickness were the two most crucial aspects determining dimensional accuracy, which was conditioned by the length of the movement of the extruder, and the accumulation of layers:(a)With reference to the build orientations, only the on-edge orientation obtained good dimensional results in all three axes, with dimensional deviations below ~ 13 µm, which was a very satisfactory rate. The optimum conditions were obtained for a layer thickness of 0.06 mm, and a feed rate values ranging from 20 mm/s to 50 mm/s.(b)For the X-Y axes, the lowest dimensional variability was obtained in the upright orientation on the X-Y axes, with the shortest movement in length of the extruder. For the Z axis, the lowest dimensional variability was observed in the flat orientation, with a smaller accumulation of layers.(c)Though feed rate had no significant effect on any of the build orientations, and the effect of layer thickness failed to produce a clear behavioral pattern, most of the optimum conditions were obtained for *Fr* ≤ 50 mm/s and *Lt* ≤ 0.12 mm.

The “bathtub ring” and waviness effects significantly increased the flatness error. The “bathtub ring” was due to excess depositing of material in the extruder acceleration and deceleration areas. This effect was heightened with increased feed rate, and reached a maximum with the highest feed rate of 80 mm/s. Waviness was due to defects in the spindle rotation of the build plate, and in the guides of the extruder. The lowest flatness values were obtained in the upright orientation with values of 7.8 µm with a feed rate of 50 mm/s. 

Surface roughness exhibited two different behaviors that depended on the build orientation. The upright and on-edge positions showed a linear increase in *Sa* and *Sz* when layer thickness was increased, but the feed rate had no significant effect. The minimum values obtained for *Sa* in these orientations ranged from ~2.5 µm to ~2.8 µm, obtained with a 0.06 mm layer thickness. In the flat orientation, surface roughness was conditioned by nozzle diameter, leading to a sharp increase in surface roughness with minimum *Sa* value of ~25 µm. 

In short, the geometric behavior of PLA polymer processing by FFF additive manufacturing has been established. The results showed that excellent geometric qualities could be obtained in PLA polymer processing by FFF if the optimum conditions are used. The functional products can be manufactured by FFF with geometric qualities similar to those obtained by conventional manufacturing processes. 

## Figures and Tables

**Figure 1 polymers-11-01581-f001:**
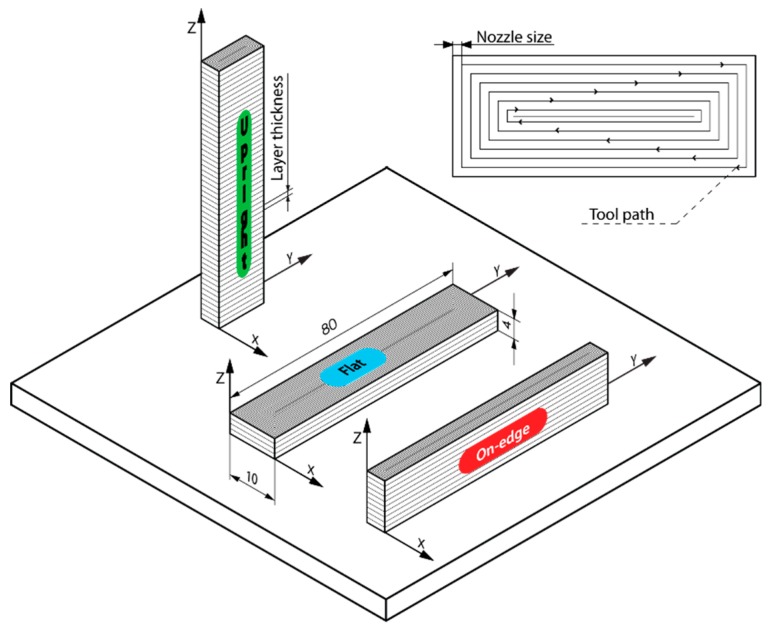
Detail of workpiece geometry and build orientations (on-edge, flat, and upright).

**Figure 2 polymers-11-01581-f002:**
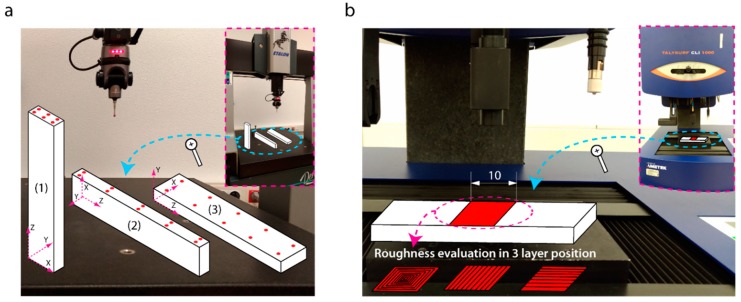
Experimental measurement setup: (**a**) Detail of the dimensional deviation measurement procedure for upright orientation workpiece; (**b**) detail of the flatness, topography, and roughness evaluation procedure for the three building orientations.

**Figure 3 polymers-11-01581-f003:**
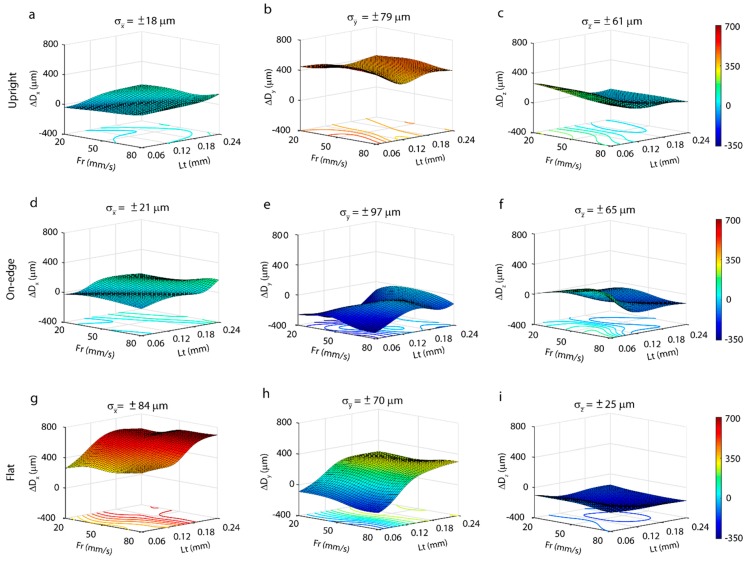
Characterization of dimensional deviation of polylactic acid (PLA) polymer processed by fused filament fabrication (FFF) for the three building orientations: Upright (**a**–**c**), on-edge (**d**–**f**), and flat (**g**–**i**).

**Figure 4 polymers-11-01581-f004:**
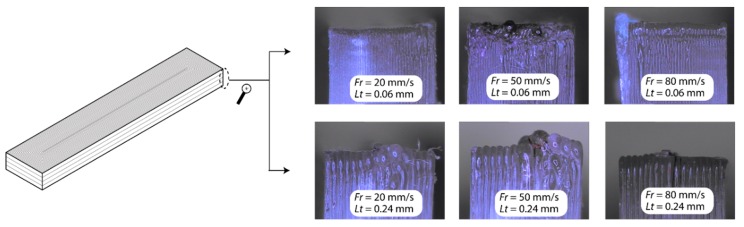
Malformation produced in the cured layer in the flat orientation.

**Figure 5 polymers-11-01581-f005:**
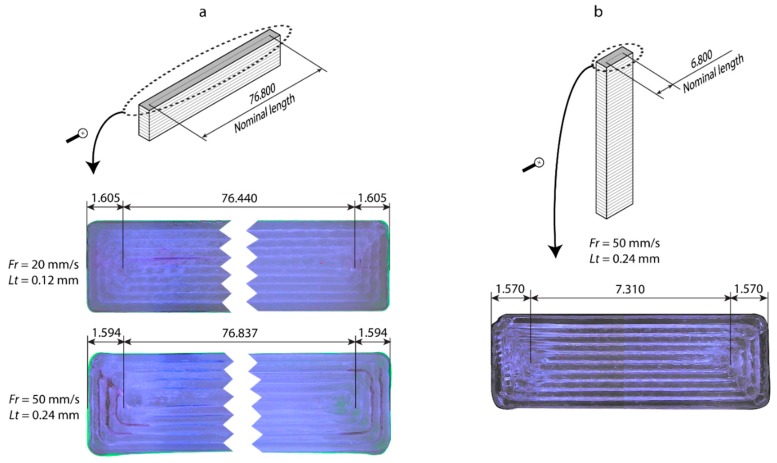
Details of the incorrect length of the central filament on the upper layer: (**a**) Flat and (**b**) upright orientation.

**Figure 6 polymers-11-01581-f006:**
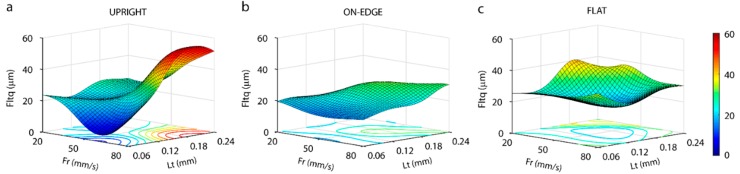
Response surface obtained by artificial neural networks for flatness in the three build orientations: (**a**) Upright, (**b**) on-edge and (**c**) flat.

**Figure 7 polymers-11-01581-f007:**
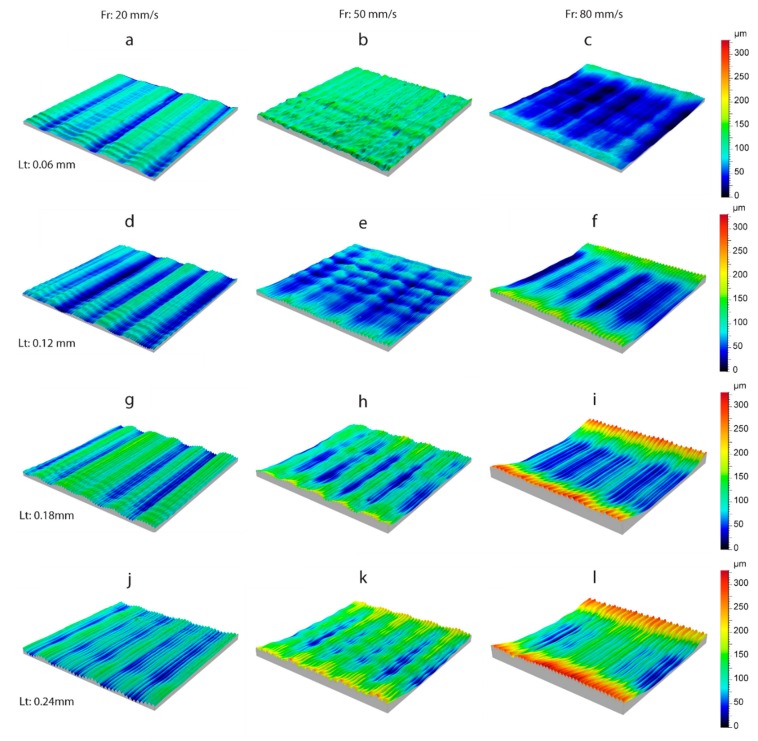
Detail of surface 3D topography for the upright orientation: (**a**) *Lt* = 0.06 mm, *Fr* = 20 mm/s; (**b**) *Lt* = 0.06 mm, *Fr* = 50 mm/s; (**c**) *Lt* = 0.06 mm, *Fr* = 80 mm/s; (**d**) *Lt* = 0.12 mm, *Fr* = 20 mm/s; (**e**) *Lt* = 0.12 mm, *Fr* = 50 mm/s; (**f**) *Lt* = 0.12 mm, *Fr* = 80 mm/s; (**g**) *Lt* = 0.18 mm, *Fr* = 20 mm/s; (**h**) *Lt* = 0.18 mm, *Fr* = 50 mm/s; (**i**) *Lt* = 0.18 mm, *Fr* = 80 mm/s; (**j**) *Lt* = 0.24 mm, *Fr* = 20 mm/s; (**k**) *Lt* = 0.24 mm, *Fr* = 50 mm/s; (**l**) *Lt* = 0.24 mm, *Fr* = 80 mm/s.

**Figure 8 polymers-11-01581-f008:**
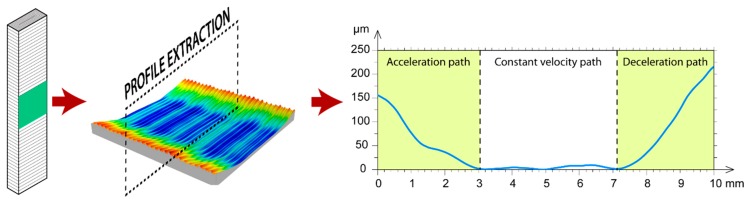
Detail of profile extraction in an upright orientation.

**Figure 9 polymers-11-01581-f009:**
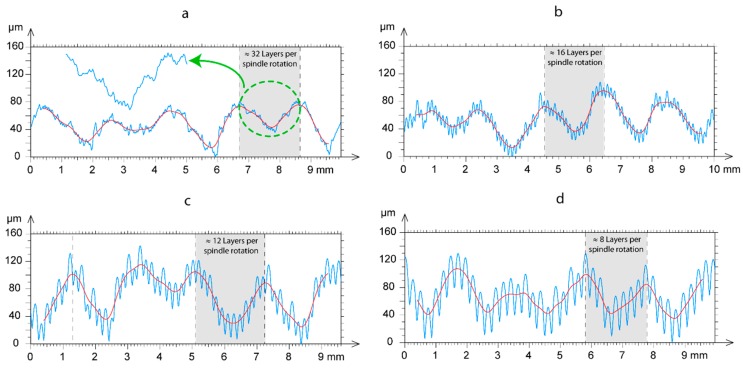
Detail of periodic waviness in an upright orientation: (**a**) *Lt* = 0.06 mm, (**b**) *Lt* = 0.12 mm, (**c**) *Lt* = 0.18 mm and (**d**) *Lt* = 0.24 mm.

**Figure 10 polymers-11-01581-f010:**
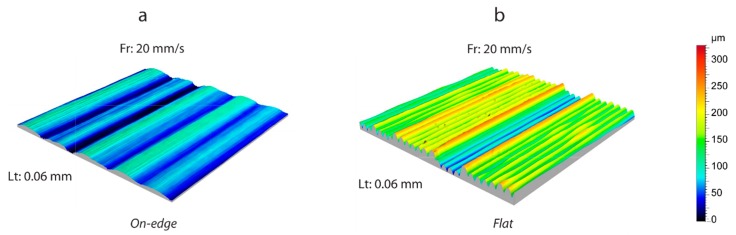
Detail of surface 3D topography: (**a**) On-edge, and (**b**) flat.

**Figure 11 polymers-11-01581-f011:**
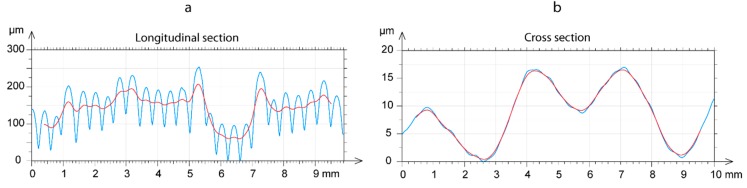
Detail of profile extraction in flat orientation: (**a**) Longitudinal and (**b**) cross-section.

**Figure 12 polymers-11-01581-f012:**
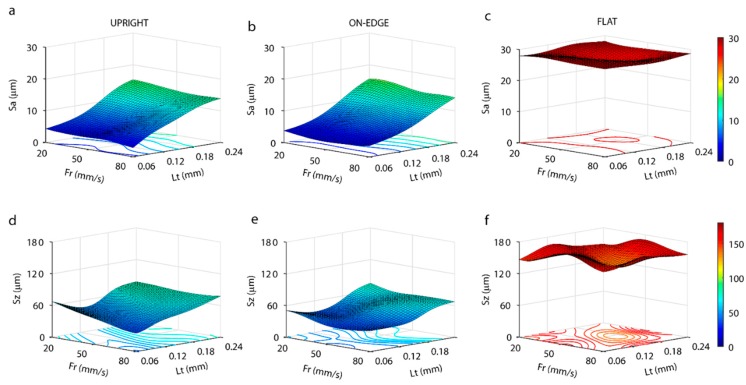
Response surface obtained by artificial neural networks for roughness: *Sa* parameter (**a**–**c**), and *Sz* parameter (**d**–**f**).

**Figure 13 polymers-11-01581-f013:**
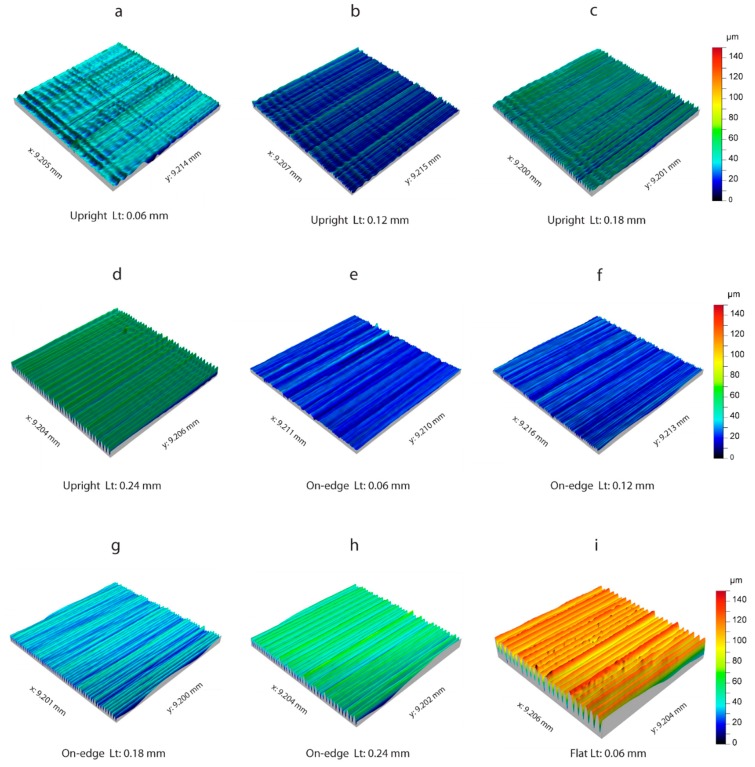
Detail of surface roughness obtained in the three build orientations: Upright (**a**–**d**), on-edge (**e**–**h**), and flat (**i**).

**Table 1 polymers-11-01581-t001:** ANOVA for dimensional deviations.

Parameter	Build Orientation					
	Upright		On-Edge		Flat	
	Sum of Squares	*p*-Value	Sum of Squares	*p*-Value	Sum of Squares	*p*-Value
Fr	477	0.946	19843	0.323	23	0.981
Lt	36777	0.550	3669	0.669	20819	0.472
$Fr^{2}$	26	0.987	18241	0.343	10	0.987
$Lt^{2}$	15960	0.693	6615	0.566	16087	0.526
Fr×Lt	1833	0.893	71	0.953	500	0.911

**Table 2 polymers-11-01581-t002:** Pearson´s correlation for dimensional accuracy.

Parameter	Build Orientation		
	Upright	On-Edge	Flat
*Fr*	−0.007	0.107	0.002
*Lt*	0.295	0.091	−0.106

**Table 3 polymers-11-01581-t003:** ANOVA analysis of flatness

Parameter	Build Orientation					
	Upright		On-Edge		Flat	
	Sum of Squares	*p*-Value	Sum of Squares	*p*-Value	Sum of Squares	*p*-Value
Fr	481.48	0.004	13.006	0.388	0.089	0.967
Lt	797.95	0.001	84.929	0.055	15.144	0.591
$Fr^{2}$	250.05	0.017	5.737	0.559	59.844	0.302
$Lt^{2}$	56.68	0.171	19.836	0.294	58.224	0.308
Fr×Lt	329.13	0.010	35.939	0.173	28.143	0.468

**Table 4 polymers-11-01581-t004:** ANOVA surface roughness.

Area Roughness Parameters	Processing Parameters	Build Orientation					
		Upright		On-Edge		Flat	
		Sum of Squares	*p*-Value	Sum of Squares	*P*-Value	Sum of Squares	*p*-Value
*Sa*	*Fr*	0.058	0.809	0.726	0.157	0.03251	0.849
	*Lt*	201.044	0.000	225.816	0.000	0.00726	0.928
	$Fr^{2}$	1.402	0.259	0.072	0.630	1.70134	0.200
	$Lt^{2}$	0.261	0.610	3.101	0.016	1.12853	0.285
	Fr×Lt	0.019	0.888	0.477	0.238	0.00342	0.951
*Sz*	*Fr*	123.48	0.182	31.96	0.436	26.21	0.736
	*Lt*	1338.88	0.003	1535.81	0.001	0.85	0.951
	$Fr^{2}$	0.11	0.965	0.07	0.971	2.04	0.925
	$Lt^{2}$	108.90	0.206	106.56	0.178	25.75	0.739
	Fr×Lt	206.25	0.099	4.17	0.773	8.89	0.844

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
