# Peer review of "Analysis of PLA Geometric Properties Processed by FFF Additive Manufacturing: Effects of Process Parameters and Plate-Extruder Precision Motion"

_polymers, 2019, doi:10.3390/polym11101581_

Round 1

Reviewer 1 Report

This paper studied the additive manufacturing of PLA and how the processing influence the structural features. 

the paper is too long and will easily lose the readers' attension. e.g., the introduction is very informative but it shows the lack of concentration. what are the main challenges in FDM PLA? or similar polymers? how does this research solve those problems? list of many literature studies and how they did the research did not provide addon values. Similarly, the characterizations or modeling should be concise enough to give the 'selling points'. other information Can be either eliminated or put in the supporting information in case the readers are interested.  e.g., the list of all ANOVA data does not provide straight information about how the printing method influences the surface morphology. The conclusion is the same - it is too long and the main contribution from this work is just diluted.  The processing influence on the material structure is indeed obvious -what is more important is, what is the author's conclusion regarding the structure control for property enhancement? any tests results to approve the authors' conclusion?

Author Response

RESPONSE TO REVIEWER 1 COMMENTS

We thank the reviewers for their constructive criticism and for pointing out issues to improve the paper. We have found them very useful and we have implemented changes and clarifications in the revised manuscript accordingly. All new or modified text in the revised manuscript has been highlighted using red background. Below is our detailed response to the reviewers’ comments:

English language and style

( ) Extensive editing of English language and style required 
( ) Moderate English changes required 
(x) English language and style are fine/minor spell check required 
( ) I don't feel qualified to judge about the English language and style 

Response: In order to improve the standard of English, the manuscript has been revised by a native professional and qualified translator, Mr. Romen Das Gupta, who is a registered member of the Spanish Association of Translator (ASETRAD), membership number 624.

Comment 1: This paper studied the additive manufacturing of PLA and how the processing influence the structural features. The paper is too long and will easily lose the readers' attension. e.g., the introduction is very informative but it shows the lack of concentration. What are the main challenges in FDM PLA? or similar polymers? how does this research solve those problems? list of many literature studies and how they did the research did not provide addon values.

Response 1: According to the Reviewer´s suggestion, the whole manuscript has been revised in order to eliminate parts of the paper with superfluous information. The introduction has been shortened by almost one page, eliminating unnecessary information and determining the current main challenges in FFF processes.

Comment 2: Similarly, the characterizations or modeling should be concise enough to give the 'selling points'. other information Can be either eliminated or put in the supporting information in case the readers are interested.  e.g., the list of all ANOVA data does not provide straight information about how the printing method influences the surface morphology.

Response 2: The authors agree with the Reviewer´s comment. The comments on the evaluation of the influence of 3D printing parameters on dimensional accuracy have been synthesized. The analysis of dimensional relative errors has been removed in order to clarify the final results and give the “selling points”. Regarding the ANOVA, the authors consider this information is essential for the understanding of the manuscript as this analysis determines the significance of 3D printing parameters in relation to dimensional accuracy, flatness, and surface roughness. Furthermore, both the ANOVA and Pearson study allow us to establish the linearity and non-linearity of the printing parameters, so this analysis is necessary to justify the use of artificial neural networks for building the surface responses. However, if the Reviewer considers it essential to eliminate this analysis, we will remove it from the manuscript.

Comment 3: The conclusion is the same - it is too long and the main contribution from this work is just diluted.  The processing influence on the material structure is indeed obvious -what is more important is, what is the author's conclusion regarding the structure control for property enhancement? any tests results to approve the authors' conclusion?

Response 3: According to the Reviewer´s suggestion, the conclusions have been simplified and shortened in order to provide readers a clearer and concise overview of this study. The conclusions have focused on determining the optimum 3D printing parameters to obtain significant enhancements in the control of dimensional and surface texture properties in FFF process with PLA material. Regarding the structure control for property enhancement, this analysis is outside the objective of this manuscript. However, the study of mechanical properties of this material in FFA processes was published in 2017 in the journal Material and Design [5].  The results shown in the conclusion have been verified with the experimental measurement of dimensional accuracy, flatness, and surface roughness of the manufactured parts.

[5] Chacón, J.M.; Caminero, M.A.; García-Plaza, E.; Núñez, P.J. Additive manufacturing of PLA structures using fused deposition modelling: effect of process parameters on mechanical properties and their optimal selection. Mater. Des. 2017, 124, 143–157.

Reviewer 2 Report

The article "Analysis of geometric properties of Polylactic Acid (PLA) processed by FFF additive manufacturing: effects of process parameters and plate-extruder precision motion" is well written and sound in technical side. I suggest publication with major revision.

First few sentences of abstract is too generic, hence can be shortened or omitted.

Citation is not in right form, like [25] [26] [22], should be [22, 25, 26].

The ANOVA should be referred to right source. For instance, the article in ‘Measurement 121, 249-260’.

ANN parameters should be compared with literature, and justified their selection too. For instance with Measurement 92, 464-474.

Comparison of your findings with that of literature is mostly missing. Should be done extensively. It would be nice to see the comparison of the results with the article "Optimization and reliability analysis to improve surface quality and mechanical characteristics of heat-treated fused filament fabricated parts". 

Moreover, the conclusion should be more refined - presentation of what is done, what is found and implication of the found results.

Author Response

RESPONSE TO REVIEWER 2 COMMENTS

The article "Analysis of geometric properties of Polylactic Acid (PLA) processed by FFF additive manufacturing: effects of process parameters and plate-extruder precision motion" is well written and sound in technical side. I suggest publication with major revision.

In order to improve the standard of English, the manuscript has been revised by a native professional and qualified translator, Mr. Romen Das Gupta, who is a registered member of the Spanish Association of Translator (ASETRAD), membership number 624.

We thank the reviewers for their constructive criticism and for pointing out issues to improve the paper. We have found them very useful and we have implemented changes and clarifications in the revised manuscript accordingly. All new or modified text in the revised manuscript has been highlighted using red background. Below is our detailed response to the reviewers’ comments:

Comment 1: First few sentences of abstract is too generic, hence can be shortened or omitted.

Response 1: According to the Reviewer´s suggestion, the first sentences of the abstract have been deleted.

Comment 2: Citation is not in right form, like [25] [26] [22], should be [22, 25, 26].

Response 2: The authors agree with the Reviewer´s comment. The citations have been rewritten in the correct form.

Comment 3: The ANOVA should be referred to right source. For instance, the article in ‘Measurement 121, 249-260’.

Response 3: The authors agree with the Reviewer´s suggestion. The proposed reference is ideal for describing the ANOVA analysis and it has been added to the manuscript.

Comment 4: ANN parameters should be compared with literature, and justified their selection too. For instance with Measurement 92, 464-474.

Response 4: According to the Reviewer´s suggestion, the ANN parameters have been compared with the literature. The proposed article has been added to the manuscript.

Comment 5: Comparison of your findings with that of literature is mostly missing. Should be done extensively. It would be nice to see the comparison of the results with the article "Optimization and reliability analysis to improve surface quality and mechanical characteristics of heat-treated fused filament fabricated parts". 

Response 5: The authors agree with the Reviewer´s suggestion. New section has been added in the manuscript to establish a comparison between our results and the literature. Authors have found most interesting the article recommended by the Reviewer. This work examines the effect of heat treatment in ABS parts in terms of several aspects such as surface roughness, hardness, dimensional accuracy, tensile strength, flexural strength, and impact strength. The parts are fabricated by FFF using fixed 3D printer parameters such as orientation (0°), layer thickness (0.3 mm), raster angle (± 45°), raster width (0.5 mm), and printing speed (35 mm/s), with tree infill densities. The proposed work has been added in the introductory section to explain how surface roughness can be improved in FFF processes. However, the authors consider this study is not directly related to the present work, being impossible to establish a real comparison, so this manuscript has not been included in the comparative section.    

Comment 6: Moreover, the conclusion should be more refined - presentation of what is done, what is found and implication of the found results.

Response 6: According to the Reviewer´s suggestion, the conclusions have been abbreviated and improved. In addition, this section has been shortened to clarify the main contribution of this work.

Round 2

Reviewer 1 Report

accept

Reviewer 2 Report

No more comments.